# Label-Free Human Disease Characterization through Circulating Cell-Free DNA Analysis Using Raman Spectroscopy

**DOI:** 10.3390/ijms241512384

**Published:** 2023-08-03

**Authors:** Vassilis M. Papadakis, Christina Cheimonidi, Maria Panagopoulou, Makrina Karaglani, Paraskevi Apalaki, Klytaimnistra Katsara, George Kenanakis, Theodosis Theodosiou, Theodoros C. Constantinidis, Kalliopi Stratigi, Ekaterini Chatzaki

**Affiliations:** 1Institute of Molecular Biology and Biotechnology, Foundation for Research and Technology, 70013 Heraklion, Greece; vassilis_papadakis@imbb.forth.gr (V.M.P.); chris_cheimonidi@imbb.forth.gr (C.C.); bio2278apalaki@gmail.com (P.A.); callina@imbb.forth.gr (K.S.); 2Department of Industrial Design and Production Engineering, University of West Attica, 12244 Athens, Greece; 3Institute of Agri-Food and Life Sciences, University Research & Innovation Center, Hellenic Mediterranean University, 71410 Heraklion, Greece; mpanagop@med.duth.gr (M.P.); mkaragla@med.duth.gr (M.K.); 4Laboratory of Pharmacology, Medical School, Democritus University of Thrace, 68100 Alexandroupolis, Greece; theodosios.theodosiou@gmail.com; 5Institute of Electronic Structure and Laser, Foundation for Research and Technology-Hellas, N. Plastira 100, Vasilika Vouton, 70013 Heraklion, Greecegkenanak@iesl.forth.gr (G.K.); 6Department of Agriculture, Hellenic Mediterranean University—Hellas, Estavromenos, 71410 Heraklion, Greece; 7Laboratory of Hygiene and Environmental Protection, Medical School, Democritus University of Thrace, 68100 Alexandroupolis, Greece; tconstan@med.duth.gr

**Keywords:** cell-free DNA, liquid biopsy, Raman spectroscopy, biomolecular, human, cancer, diabetes

## Abstract

Circulating cell-free DNA (ccfDNA) is a liquid biopsy biomaterial attracting significant attention for the implementation of precision medicine diagnostics. Deeper knowledge related to its structure and biology would enable the development of such applications. In this study, we employed Raman spectroscopy to unravel the biomolecular profile of human ccfDNA in health and disease. We established reference Raman spectra of ccfDNA samples from healthy males and females with different conditions, including cancer and diabetes, extracting information about their chemical composition. Comparative observations showed a distinct spectral pattern in ccfDNA from breast cancer patients taking neoadjuvant therapy. Raman analysis of ccfDNA from healthy, prediabetic, and diabetic males uncovered some differences in their biomolecular fingerprints. We also studied ccfDNA released from human benign and cancer cell lines and compared it to their respective gDNA, confirming it mirrors its cellular origin. Overall, we explored for the first time Raman spectroscopy in the study of ccfDNA and provided spectra of samples from different sources. Our findings introduce Raman spectroscopy as a new approach to implementing liquid biopsy diagnostics worthy of further elaboration.

## 1. Introduction

In the search for circulating biomarkers to enable accurate disease screening, diagnosis, prognosis, monitoring, and therapy response assessment, increasing interest has been attracted over the last decade to circulating cell-free DNA (ccfDNA). CcfDNA is a liquid biopsy biomaterial believed to be enriched by pathological tissues or tumors and, therefore, is able to mirror their genetic and epigenetic condition [1]. CcfDNA was identified in the bloodstream by Mendel and Métais in 1948 [2]; however, it was 30 years later that ccfDNA concentration was found higher in cancer patients than healthy individuals [3], leading the way for many studies to elucidate the actual nature of this biological entity and evaluate its validity as a clinically useful source for biomarker identification. Among others, we have studied ccfDNA biology in vitro [4] and in breast cancer patients [5], where ccfDNA emerged as a highly potent predictive classifier in metastatic breast cancer [6]. Tissue-specific cell-free DNA degradation seems to be the key to quantifying the burden of ccfDNA released by tumors [7], whereas many studies have investigated the potential of ccfDNA to become a blood-based cancer screening tool [8]. CcfDNA analysis is also important in minimally invasive prenatal testing [6,7,8,9], organ transplantation [10], and the detection of immune diseases, such as active lupus [11] and psoriasis [12]. Moreover, lately, studies on diabetes have revealed that epigenetic modifications in ccfDNA can serve as biomarkers of beta cell death [13,14,15,16].

Raman spectroscopy is a state-of-the-art vibrational spectroscopy technique that is based on the analysis of scattered light, enabling the observation and detection of the transition between the different vibrational energy states of a molecule [17]. Raman spectra reveal the chemical composition of measured samples to bring out their unique biomolecular fingerprint. Raman spectroscopy has become very attractive for life sciences since it is label-free, highly detailed, and does not require pretreatment, with minimal operational costs [18]. In addition, the analysis of cells and tissues is nondisruptive, especially from water molecules, and it can be quantified and performed in situ [19]. Moreover, recently established techniques for the enhancement of Raman scattering can increase the signal-to-noise ratio with the use of nanoparticles, such as surface-enhanced Raman scattering [20,21,22], or laser excitation frequency, such as resonance Raman scattering [23]. Different biomolecules have been analyzed with the use of Raman spectroscopy, such as chromosomes and single living cells [24], extracellular matrix [25], human blood cells [26,27,28], lymphocytes [29], mammalian cell cultures [30], and cancer cells but also tissues, such as cervical tissue [31] and epithelial tissue [32]. A Raman endoscope was also developed to study colorectal cancer in a mouse model [33]. Serum and plasma components have been analyzed before with the use of Raman spectroscopy [34] and nucleic acids as well [35]; nevertheless, ccfDNA has not been studied so far.

In this study, we aimed to fill this gap; we sought to explore deeper the biological characteristics of ccfDNA with the use of Raman spectroscopy, reveal its biomolecular fingerprint in health and disease, and seek information that could be of clinical value. We investigated for the first time human ccfDNA from healthy males and females, mammary cell lines, and selected representative pathological conditions of major burden: one malignancy (breast cancer) and one metabolic (type 2 diabetes mellitus-T2D). Original spectra were produced and analyzed to reveal important basic information on ccfDNA biomolecular patterns and differences between study groups, as well as gDNA.

## 2. Results

### 2.1. Raman Spectra of ccfDNA from Healthy Individuals Share Common Patterns Indicative of Their Biomolecular Profiles

In order to describe the biomolecular profile of ccfDNA in health, we first examined samples from male and female healthy individuals via Raman spectroscopy.

Raman spectrum analysis of ccfDNA from three male and three age-matched female healthy individuals revealed 11 peaks (Figure 1). Measurements were tested for repeatability across samples by acquiring spectra from five points from each sample, as described in the methodology section. Results showed that Raman spectra were similar, with no significant differences between measurements in the same sample. Regarding differences between samples, as can be seen in Appendix A, there is a higher inhomogeneity between the male subjects compared to the female subjects. The SD of the male subjects is significant in the range of 800–1000 cm^−1^ and 2800–3000 cm^−1^, while the SD of female subjects is very close to the noise level.

In Table 1, the Raman peaks identified in the spectra of all ccfDNA samples investigated in this study are listed with their corresponding assignments from the literature. Healthy male and female ccfDNA spectra both showed nine common major peaks at 525–529 cm^−1^ [36,37], 758–762 cm^−1^ [38], 884 cm^−1^ [38], 1002–1004 cm^−1^ [39,40], 1046–1052 cm^−1^ [41], 1345–1348 cm^−1^ [42], 1460 cm^−1^ [39], 2889–2891 cm^−1^ [43], and 2942–2944 cm^−1^ [43]. However, interestingly, a few differences were identified between genders, shown in all studied samples: female samples showed a unique peak at 914 cm^−1^ [36,44], while male samples had a unique peak at 963 cm^−1^ [36].

Overall, we describe for the first time the biomolecular profile of ccfDNA in health through Raman spectroscopy, producing distinct spectra patterns of samples from healthy subjects.

### 2.2. Raman Spectra of ccfDNA and gDNA from Human Normal and Cancer Breast Cell Lines

To gain more insight into the unique ccfDNA biomolecular features, we also analyzed the Raman profile of ccfDNA released by the human epithelial mammary cell line MCF12A and compared it to the cells’ genomic DNA (Figure 2). Spectra analysis revealed 13 major peaks, all present both in ccfDNA and gDNA listed in Table 1 with their corresponding assignments. These peaks were present at 527–529 cm^−1^ [36,37], 621–624 cm^−1^ [37], 760–765 cm^−1^ [38], 885–887 cm^−1^ [38], 1005 cm^−1^ [39,40], 1063–1065 cm^−1^ [42], 1130–1138 cm^−1^ [36], 1223–1246 cm^−1^ [38], 1463 cm^−1^ [39], 1552–1553 cm^−1^ [40], 1574–1578 cm^−1^ [51,52,53], 2889–2891 cm^−1^ [43], and 2942–2944 cm^−1^ [43]. As observed, all ccfDNA spectra from healthy female samples so far share eight common peaks that may be indicative of the biomolecular profile of ccfDNA. Most importantly, these results show that ccfDNA bears the same traits and biological properties as the gDNA from its cellular origin. Raman spectra within the region (1800–2800 cm^−1^) belong to the silent region, in which no strong signals are derived from endogenous biomolecules. Within this silent region, some functional groups present low signals, and particularly, in our case, the alkyne group has a (–C≡C– stretching vibration) at 2110 cm^−1^ [54].

We then investigated the profiles of ccfDNA and genomic DNA from the breast cancer cell line MCF-7 (Figure 3). Analysis found 13 major peaks, all present both in ccfDNA and respective gDNA that are listed in Table 1 with their corresponding assignments. These peaks are present at 526–531 cm^−1^ [36,37], 762–767 cm^−1^ [38], 781–788 cm^−1^ [38], 885–887 cm^−1^ [38], 913–921cm^−1^ [36,44], 1005 cm^−1^ [39,40], 1063–1065 cm^−1^ [42], 1223–1246 cm^−1^ [51], 1460–1462 cm^−1^ [39], 1552–1553 cm^−1^ [43], 1650 cm^−1^ [50], 2889–2891 cm^−1^ [43], and 2942–2944 cm^−1^ [43]. From those, 10 are also found in the MCF12A normal mammary cell line, while 9 are common between healthy female samples. Similar spectra between MCF-7 ccfDNA and gDNA again confirm that ccfDNA mirrors the gDNA of its cellular origin.

### 2.3. Raman Spectroscopy Reveals Changes in ccfDNA Biomolecular Profile in Breast Cancer Patients

We then continued with an analysis of ccfDNA from patients suffering from breast cancer. Three groups of patients were studied (Appendix A): patients receiving adjuvant chemotherapy (n = 7), patients with metastasis (n = 8), and patients receiving neoadjuvant therapy (n = 5). Average Raman spectra are shown in Figure 4a, Figure 4b, and Figure 4c, respectively, where the analysis revealed overall 11 major peaks (Figure 4d) described along their corresponding assignments in Table 1. Overall, spectra presented a repeatable profile within the same patient group and were found similar to those from healthy individuals; however, some significant differences emerged in the neoadjuvant breast cancer patients. Six identified peaks were found to be present in all healthy and breast cancer groups at 529–536 cm^−1^ [36,37], 758–769 cm^−1^ [38], 1002–1008 cm^−1^ [39,40], 1456–1464 cm^−1^ [39], 2889–2891 cm^−1^ [43], and 2942–2944 cm^−1^ [43] described according to their assignments in Table 1. There is one peak at 1345 cm^−1^ [42] corresponding to a C–N stretching vibration of thymine, which is only present in healthy female ccfDNA samples and not in any of the breast cancer patients. The most exciting finding, however, was that ccfDNA from breast cancer patients taking neoadjuvant therapy showed four unique peaks at 725 cm^−1^ [46], 1302 cm^−1^ [47], 1336 cm^−1^ [48], and 1577 cm^−1^ [51,52], presenting a specific pattern distinct from all other patient groups, with the 1577 cm^−1^ also found in healthy groups and the last also found in MCF12A cells.

These interesting results led to the next series of measurements that engaged sequential samples from the same breast cancer patient. There were two sets of samples from four metastatic and two adjuvant breast cancer patients. Sampling was performed before initiation of therapy and 3 to 6 months later in order to examine whether Raman analysis can detect dynamic changes in ccfDNA during the course of the disease and treatment. Raman spectra from a representative adjuvant breast cancer patient and from a metastatic patient are shown in Figure 5, respectively. The 13 identified major peaks and their corresponding assignments are presented in Table 1. All of the identified peaks at 522 cm^−1^ [36,37], 740 cm^−1^ [38], 921 cm^−1^ [36,44], 1003 cm^−1^ [39,40], 1101 cm^−1^ [46], 1223 cm^−1^ [38], 1372 cm^−1^ [49], 1398 cm^−1^ [50], 1456–1464 cm^−1^ [39], 1545 cm^−1^ [40], 1603 cm^−1^ [53], 2889–2891 cm^−1^ [43], and 2942–2944 cm^−1^ [43] were found to be present in all patients. No differences were observed between samples from the same patient.

### 2.4. Raman Spectroscopy Reveals Differences in the ccfDNA Biomolecular Profile of Prediabetic and Diabetic Males

In the following part of the study, ccfDNA samples from prediabetic and T2D male patients were analyzed through Raman Spectroscopy, and spectra were compared with those from healthy male individuals to reveal differences in their biomolecular profile during this metabolic pathology development. We observed repeatable spectral profiles within the same patient group. Raman spectrum analysis revealed eight main peaks in the pathological samples with their corresponding assignments presented in Table 1. Two peaks detected in the healthy individuals, 1046 cm^−1^ [42] and 1348 cm^−1^ [42], were absent in both diabetic and prediabetic groups (Figure 6). Shared peaks were found at 526–529 cm^−1^ [36,37], 762 cm^−1^ [38], 882–884 cm^−1^ [38], 1003–1006 cm^−1^ [39,40], 1460–1462 cm^−1^ [39], 2883–2891 cm^−1^ [43], and 2942–2948 cm^−1^ [43]. Healthy and diabetic ccfDNA samples also share a common peak at 963 cm^−1^, absent in the prediabetic samples.

## 3. Discussion

Raman spectroscopy is a well-established spectroscopic method that gained a lot of attention in the last years, mainly because of the recent technological advancements in lasers and optics. It is sensitive, selective, and noninvasive, while it is not limited to water absorption, allowing the study of biological samples. Εvery biological molecule has a unique vibrational pattern that can serve as a marker and potentially be used for clinical diagnosis in the future. CcfDNA has emerged as a liquid biopsy material attracting increasing interest in the field of molecular diagnostics, and multiple approaches have been employed to characterize its quantitative and qualitative profile in order to understand its biology and enable clinical applications. Raman spectroscopy has been used before to study different biochemical compounds, such as nucleic acids [35], proteins and lipids [55], cellular components such as nuclei and mitochondria [37,53], and blood and the tissues’ morphological structures [34,56,57,58]. However, ccfDNA has not been investigated before with Raman spectroscopy to reveal its biomolecular content, which can enrich our knowledge of its biology. For example, the detection of methyl groups by Raman spectroscopy reflects global methylation load being very important in gene expression regulation both in physiological and pathological conditions. In this study, we have demonstrated that Raman spectroscopy can reveal valuable information for the biomolecular profiling of ccfDNA in health and pathology, potentially with clinical utility.

We studied ccfDNA samples from healthy individuals, breast cancer patients, prediabetic and diabetic patients, and human healthy and cancer cell lines and provided, for the first time, the respective reference spectra and distinct patterns for this biomaterial. Spectra share many peaks indicative of all four nucleotide bases A, G, T, C, the phosphate backbone of DNA, and carbon asymmetric stretching that is indicative of DNA methylation. Six peaks were present in all samples, forming a pattern, whereas some other chemical entities were identified in the different study groups. Spectra were generally similar between genders. A unique peak was observed in each of the male and female groups, although the inhomogeneity between the samples in the male group raised concerns about the significance of these differences, which need to be confirmed in larger groups. There is not really much information in the literature regarding structural ccfDNA differences between genders, although levels are reported to be higher in men [59]. We can only speculate that observed differences could be attributed to the Y chromosome.

In the cell lines, ccfDNA derives from a unique cell type, and therefore, direct comparisons to ccfDNA from patients are not very informative. Our results from the Raman analysis of ccfDNA from a normal and a cancer human breast cell line revealed some minor differences between the two. In the cell lines, we had the opportunity to compare ccfDNA spectra to those from the respective gDNA, to be found identical, implying that the biomolecular profile of ccfDNA mirrors the gDNA of its cellular origin. gDNA spectra from MCF12A and MCF7 cell lines were also similar between them, as we found many common peaks between the two cell lines. gDNA of human origin has not been previously studied extensively by Raman spectroscopy. Raman has been used before to identify changes in gDNA in tomato cultivars before and after cryopreservation [60] to identify gDNA in pork spleen [61] and in 3T3 mouse cells [62]. Few peaks are common in these studies and in our human gDNA data. These are at 758–769 cm^−1^ [38] and 1456–1464 cm^−1^ [39], which are common in pork spleen and in our cell lines, and at 1002–1008 cm^−1^ [39,40], 1456–1464 cm^−1^ [39], and 1552–1553 cm^−1^ [40], which are common between the mouse 3T3 cell line and our cell lines. More analyses of gDNA samples from different cell types and species may identify specific peaks and patterns.

During pathology, different mechanisms of ccfDNA release are initiated, which may result in different ccfDNA fragments or even different ratios of gDNA/mitochondrial DNA. These may result in specific patterns in the biomolecular ccfDNA profile in diseases. From a pathophysiological point of view, when comparing samples from different groups, one has to consider that in health, multiple tissues contribute with more or less ccfDNA release into the circulation pool, whereas—at least in theory—in pathology, ccfDNA is enriched by the pathological tissue. Similarly, in the case of cancer, it is considered that most ccfDNA derives from the tumor, called, therefore, circulating tumor DNA, and may reflect aneuploidy and other malignant features.

Cancer is characterized by global DNA hypomethylation [63] and we show here that Raman spectroscopy is able to detect methylation in ccfDNA and gDNA samples. This was also confirmed by Daum et al. [64], who showed that Raman microspectroscopy coupled with imaging revealed significant differences between high- and low-methylated cell types, with higher methylated cells demonstrating higher relative intensities that can be attributed to nucleobases and 5-methylcytosine. Interestingly, an enhanced SERS-based method that can directly sense the four different DNA epigenetic modifications of cytosine [62] with the use of a plasmonic gold nanohole array (PGNA) by identifying their different SERS spectral features was published recently. In our study, we were able to detect the spectral pattern of 5-methylcytosine in MCF7 cells and in breast cancer patients and also identify another peak at 760 cm^−1^ that is indicative of unmethylated cytosine. The most exciting finding, however, was that ccfDNA from neoadjuvant breast cancer patients showed four unique peaks presenting a specific pattern distinct from all other groups. These peaks are assigned mostly to purine modes (mainly G contribution), and the signal seems to be enhanced upon DNA methylation [47,53]. It is important to note that these patients still have an active primary tumor releasing tumor ccfDNA in the blood, actively enriching the ccfDNA pool. What would give an additional dimension to this approach would be the possibility to accurately quantify peaks to reveal any potential differences in sequential measurements of samples of the same patient that might be indicative of the response to treatment, suggesting the significant potential of Raman spectroscopy to monitor outcomes. This needs a different methodological approach and larger study groups in order to verify whether the signal can be accurately quantifiable to be able to indicate prognosis or response to treatment. This study is the first pilot proposing Raman spectroscopy as a potential tool for disease diagnosis or response prediction presenting advantages over current solutions. Our case study on breast cancer before and after therapy did not reveal any differences in spectra; still, the Raman approach can be studied in different settings for different cancers and clinically relevant endpoints. Larger study groups might be necessary to increase the method’s sensitivity. Furthermore, as the sample size increases, additional techniques, such as machine learning, might be incorporated, significantly improving diagnostic sensitivity, efficiency, and specificity.

Previous attempts to explore the potential of Raman spectroscopy in screening and diagnosis of human pathology focus on glucose Raman peaks from in vivo skin for noninvasive blood glucose monitoring [65,66,67]. A recent study from our group [15] demonstrates differential beta pancreatic gene methylation profiling detected in ccfDNA and a machine learning built methylation biosignature that can successfully discriminate diabetic patients from healthy individuals. In the present study, for the first time, we produce Raman spectra of ccfDNA from prediabetic and diabetic patients and compare them to those from healthy individuals to find them greatly similar. Two peaks were absent in patients, whereas healthy and diabetic ccfDNA samples share a common peak at 963 cm^−1^ absent in the prediabetic samples. This peak was identified before as vibrations from the thiolated mononucleotide of adenosine, which differs from the spectra of thiolated single-stranded DNA containing even as few as two bases [42]. Analysis of more patients will clarify whether the observed differences are specific enough to be exploited for the distinction between health and diabetes.

CcfDNA has been the focus of many studies in the last decades, presenting the opportunity of a reliable minimally invasive biomaterial for biomarker detection, valuable in clinical applications. Especially in cancer, extensive research has been performed to assess biomarkers in circulating tumor DNA, such as point mutations, copy number aberrations, chromosomal rearrangements, and fragmentomics, which can shed light on the tumor burden [7] and aid early cancer detection and cancer monitoring. Epigenetic DNA modifications, such as gene methylation relevant to tumorigenesis and cancer progression, can also be detected in ctDNA and have gained a lot of attention [68]. Moreover, ccfDNA is increasingly attracting interest in pathologies other than cancer, as it is assumed that it is released in the body fluids during necrosis and/or apoptosis from the tissue developing pathology, retaining and reflecting its epi/genetic fingerprint. In this study, we explored Raman spectroscopy as an alternative approach in the study of ccfDNA, which offers unique insights into the study of disease-related biomolecules, as also shown in this first attempt to unravel its biomolecular profile. Furthermore, the goal was to propose an alternative and potentially cost-effective and simple method to perform liquid biopsy diagnostics in humans. Results indicate that Raman spectroscopy of ccfDNA holds this potential, resulting in potentially useful diagnostic information. Our findings point to detectable differences in the Raman spectra of ccfDNAs from different sources, and the reasons behind their origin need further elaboration. The field is rather novel, however, and as we know very little about the molecular profile of ccfDNA in each condition, only speculations can be made regarding their reflection in the Raman spectra. For example, circulating tumor DNA, i.e., ccfDNA released in the blood of oncologic patients from their tumor, carries the genetic alterations occurring in the tumor, such as amplifications, deletions, rearrangements, aberrant methylation, translocations, etc., and this is probably depicted in its Raman spectrum. In the bloodstream, ccfDNA is found single-stranded, double-stranded, or in nucleoprotein complexes, with or without epi/genetic modifications. Nevertheless, more experiments should be performed in order to interpret the differences in peaks in different DNAs.

## 4. Μaterials and Methods

### 4.1. Study Groups and Clinical Samples

A total number of 31 patients was involved in this study. The study was approved by the local Scientific Boards and Ethics Committees and followed the principles of the Declaration of Helsinki. All participants took part after signing an informed consent.

Breast cancer patients were recruited in the Department of Medical Oncology of the University Hospital of Alexandroupolis, Greece. Blood samples were collected (a) from patients having recently (within the previous month) undergone surgery for primary breast cancer, exactly before the initiation of adjuvant therapy (adjuvant group), (b) from patients after diagnosis of breast cancer, having no previous surgery and before receiving chemotherapy (neoadjuvant group), and (c) from patients diagnosed with metastatic disease before the initiation of chemotherapy (metastatic group). Sequential sampling was performed before therapy initiation and six or three months after (for adjuvant and metastatic patients, respectively). Diabetic, prediabetic, and healthy blood samples were obtained from patients of the University Hospital of Alexandroupolis. T2DM and prediabetes were diagnosed according to the American Diabetes Association (ADA) guidelines [69]. Their clinical information is shown in Appendix A. All samples were of Caucasian origin.

Serum was collected within 2 h of blood sampling and was centrifuged twice (3000× *g* for 10 min, 14,000× *g* for 10 min).

### 4.2. Cell Culture

Cell culture and treatment were performed as before [4]. Briefly, the human epithelial breast cell line MCF12A and the breast cancer cell line MCF-7 were used. MCF12A cells were kindly donated from Dr V. Zoubourlis (National Hellenic Research Foundation, Greece), and they were grown in Dulbecco’s Modified Eagle’s/F12 medium (DMEM/F12, 1:1)(Biosera, Nuaille, France), with 5% equine serum (ES) (Gibco, Thermo Fisher Scientific, Waltham, MA, USA), 20 ng/mL epidermal growth factor (Peprotech, Rocky Hill, NJ, USA), 0.5 μg/mL hydrocortisone (Sigma Aldrich, MO, USA, 0.1 μg/mL cholera toxin (Sigma Aldrich, MO, USA), and 10 μg/mL insulin (Sigma Aldrich, MO, USA). MCF7 cells were grown in Dulbecco’s modified Eagle’s medium (DMEM/high glucose; Biosera, Nuaille, France), with 5% heat-inactivated fetal bovine serum (FBS; Gibco, Thermo Fisher Scientific, Waltham, MA, USA) and 1% penicillin/streptomycin (Gibco, Thermo Fisher Scientific, Waltham, MA, USA). For ccfDNA release experiments, cells were seeded into six-well plates and were incubated for 24 h in serum-free conditions. They were then washed twice with sterile phosphate-buffered saline (PBS; Hyclone, Fisher Scientific, Loughborough, UK) and were incubated in serum-free media for another 48 h. Medium was collected, and cells were collected after the use of trypsin/EDTA solution.

### 4.3. gDNA Isolation

For genomic DNA extraction from cells, the NucleoSpin Tissue Kit (Macherey Nagel, Duren, Germany) was used, as per the manufacturer’s instructions. gDNA was eluted in 50 μL elution buffer, and then it was stored at −20 °C.

### 4.4. ccfDNA Isolation

For ccfDNA extraction from serum and culture supernatants, the QIAamp Blood Mini kit (Qiagen, Hilden, Germany) was used, as per the manufacturer’s instructions. ccfDNA was eluted from 500 μL of plasma in 25 μL elution buffer, and then it was stored at −20 °C.

### 4.5. Direct Quantification of ccfDNA and gDNA

In order to quantify ccfDNA and gDNA, the Quant-iT dsDNA High-Sensitivity Assay Kit (Invitrogen, Karlsruhe, Germany) was used, and the measurement was performed in Qubit 3.0 Fluorometer (Invitrogen, Karlsruhe, Germany).

### 4.6. Raman Spectroscopy

Raman analyses took place on a LabRAM HR (Horiba, Lille, France) (Appendix A). Laser excitation was at 532 nm, while the microscope was set to acquire Raman signal in the spectral range between 300 and 3100 cm^−1^. Laser operated at 10% intensity, which resulted in 4 mW of power on the sample, and the acquisition time was set to 10 s, with 3 spectral accumulations. The selected grating was with 600 grooves, resulting in a Raman spectral resolution of around 1 cm^−1^. The objective lens used was an Olympus 50× lens with a numerical aperture (NA) of 0.5 and a working distance of 10.6 mm (LMPlanFL N, Olympus, Tokyo, Japan). A temperature-controlled stage (PE120-XY, Linkam, Surrey, UK) was coupled with the microscope stage to ensure sample’s temperature control and stability at T = 15 °C. In total, 10 μL of ccfDNA from each sample was placed on a metallic microscope plate and was left to dry (Appendix A).

Processing and analysis of the acquired raw Raman spectra were achieved through the instrument’s original software (LabSpec, version LS6, Horriba, Lille, France). Initially, cosmic rays were removed through the function Despike, and background signal was calculated by a polynomial function and subtracted from the raw Raman spectral data. No further processing (smoothing or noise cancellation processing) was performed in the Raman spectral data to ensure minimum effect in the analysis. Following Raman peaks were identified and validated through the literature. Images of dried DNA on CaF substrate are presented (Appendix A).

For statistical purposes, averaging between subject groups was performed. Typically, a minimum number of 3 subjects was selected for every category to ensure biological replication. Additionally, from each sample, 5 technical measurement points in different spatial locations were measured. The total number of Raman spectra measurements (15 in total) was averaged. Measurements were also tested for repeatability within samples.

## 5. Conclusions

In this study, for the first time, ccfDNA samples from healthy individuals and from different conditions, such as cancer and diabetes, were analyzed with Raman spectroscopy. Reference Raman spectra of ccfDNA were established, extracting information about the chemical composition, and a comparative analysis revealed biomolecular fingerprints of different sample entities, along with a distinct spectral pattern in ccfDNA from breast cancer patients taking neoadjuvant therapy. Our findings introduce Raman spectroscopy as a new approach for implementing liquid biopsy diagnostics worth further attention.

## Figures and Tables

**Figure 1 ijms-24-12384-f001:**
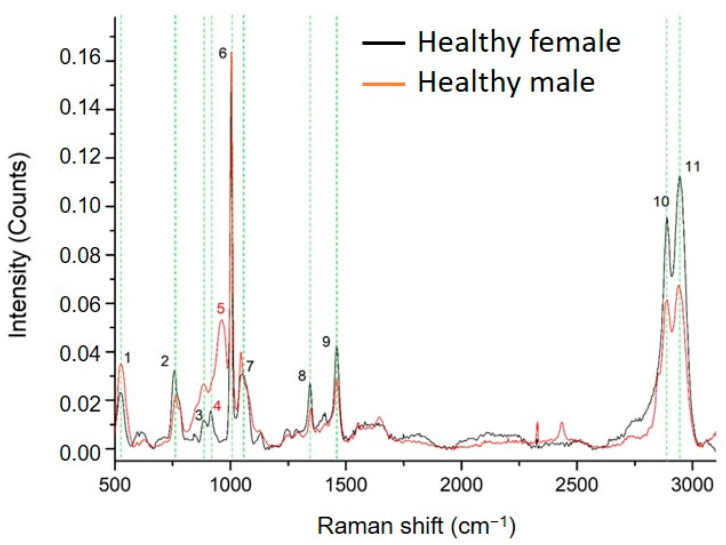
Comparison between the average Raman spectra of ccfDNA from male and female healthy individuals. The 11 major Raman peaks were identified and highlighted in green. Peak numbering in red color indicates the Raman peaks of ccfDNA that are unique in each group (Raman peak at 914 cm^−1^ is unique in female samples while 963 cm^−1^ is unique in male samples). All corresponding assignments are mentioned in the text.

**Figure 2 ijms-24-12384-f002:**
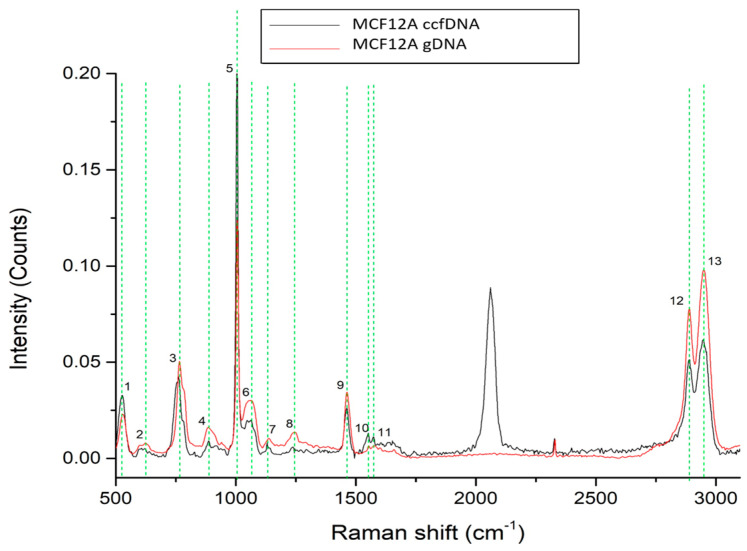
Raman spectra from gDNA and ccfDNA from the MCF12A human immortalized mammary cell line. The most distinct Raman peaks are marked in green lines. Peaks and all corresponding assignments are mentioned in the text.

**Figure 3 ijms-24-12384-f003:**
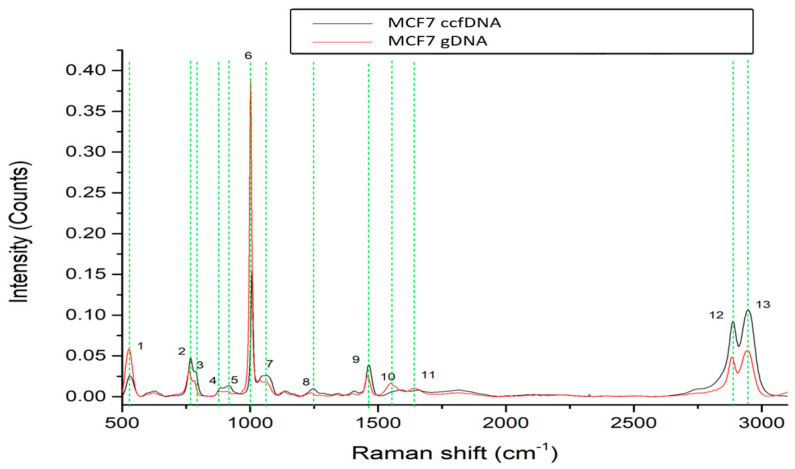
Raman spectra from gDNA and ccfDNA of the MCF7 human breast cancer cell line. The most distinct Raman peaks are marked in green lines. Peaks and all corresponding assignments are mentioned in the text.

**Figure 4 ijms-24-12384-f004:**
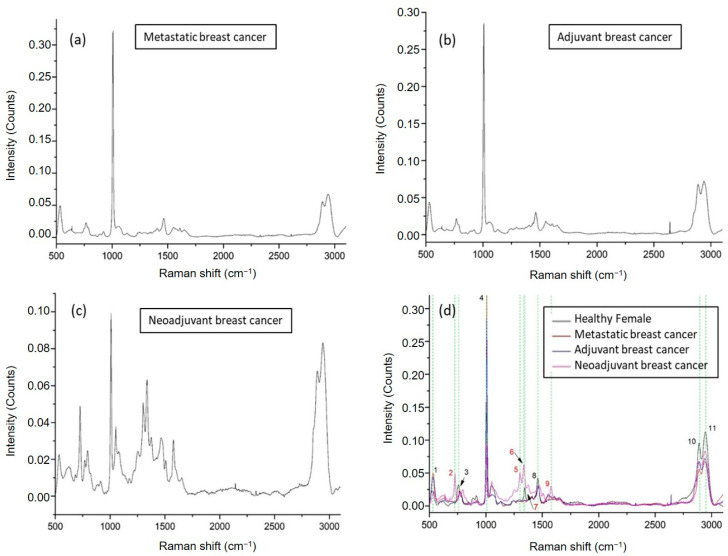
Representative Raman plots of ccfDNA samples from breast cancer patients: (**a**) spectrum from metastatic breast cancer; (**b**) from a breast cancer patient getting adjuvant therapy after surgery; (**c**) from a breast cancer patient getting neoadjuvant therapy. (**d**) Comparative Raman plot analysis of ccfDNAs between healthy individuals and different groups of breast cancer patients. The most distinct Raman peaks are marked in green lines. Peak numbering in red color indicates the Raman peaks of ccfDNA of neoadjuvant patients that are not observed in the rest of the groups. In addition, peak 7 found in female healthy ccfDNA was not present in any of the breast cancer samples. Peaks and all corresponding assignments are mentioned in the text.

**Figure 5 ijms-24-12384-f005:**
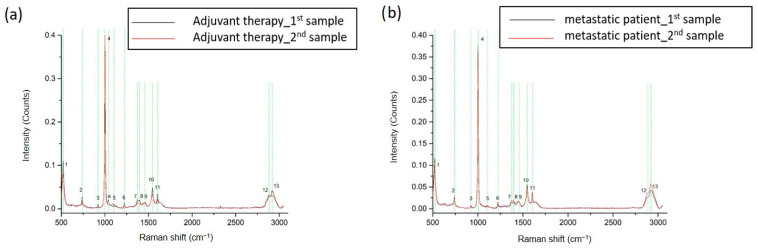
Representative Raman spectra from sequential samples of breast cancer adjuvant (**a**) and metastatic (**b**) patients before and after treatment. The most distinct Raman peaks are marked in green lines. Peaks and all corresponding assignments are mentioned in the text.

**Figure 6 ijms-24-12384-f006:**
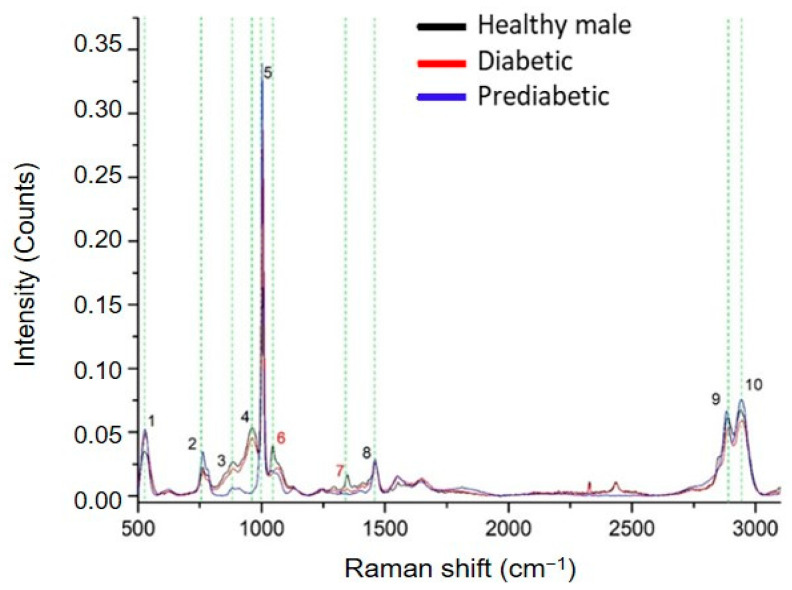
Comparative Raman plot analysis of ccfDNAs from healthy, prediabetic, and diabetic individuals. The most distinct Raman peaks are marked in green lines. Peak numbering in red color indicates peaks detected only in healthy individuals. Peaks and all corresponding assignments are mentioned in the text.

**Table 1 ijms-24-12384-t001:** Raman peaks identified in the spectra of human ccfDNA samples with their assignments, as described in the related literature.

Peak Number	Raman Shift (cm^−1^)	Assignment	Healthy Male ccfDNA	Healthy Female ccfDNA	MCF12A ccfDNA	MCF12A gDNA	MCF7 ccfDNA	MCF7 gDNA	Adj BRCA	Meta BRCA	Neoadj BRCA	T2DM Male ccfDNA	Prediabetic Male ccfDNA
(1)	522–536	524 → S-S disulfide stretching in proteins, phosphatidylserine [36] 529 → Uracil (weak) [37] 536 → Adenine [37]	√	√	√	√	√	√	√	√	√	√	√
(2)	621–624	623 → Ring breathing vibrations of A [37]			√	√							
(3)	725	726 → ^15^N–A (A = adenine) [45]									√		
(4)	740–769	746 → Ring breathing mode of thymine (T) [38]	√	√	√	√	√	√	√	√	√	√	√
(5)	780–789	780 → DNA, thymine, cytosine [38]					√	√					
(6)	879–889	883 → Five-ring deformation in the single-A spectra of the 9C1A oligonucleotide [38]	√	√	√	√	√	√				√	√
(7)	912–917	915 → Ribose vibration, one of the distinct RNA modes [36] 913 → Functional groups in genomic DNA from tomato cultivars, pontica (after (LN) plant cryopreservation) deoxyribose [44]		√			√	√					
(8)	963	963 → NH_2_ rocking vibration of thiolated A mononucleotide [36]	√									√	
(9)	1002–1009	1002 → Rocking vibration of –CH_3_ group in methylated cytosine nucleotides, [39], A, and C [40]	√	√	√	√	√	√	√	√	√	√	√
(10)	1042–1068	1050 → PO_2_^−^, symmetric stretching [42]	√	√	√	√	√	√					
(11)	1101	1101 → O–P–O backbone stretch of [46]											
(12)	1130–1138	1134 → Adenine [36]			√	√							
(13)	1223–1246	1223 → Cellular nucleic acids [38]			√	√	√	√					
(14)	1302	1302 → 5-methyl substituted of unmodified cytidine aqueous solution [47]									√		
(15)	1336	1336 → Purine bases (guanine) [48]									√		
(16)	1345–1348	1346 → C–N stretching vibration of pyrimidine of T [42]	√	√									
(17)	1372	1373 →T, A, G (ring breathing modes) [49]											
(18)	1399	1398 →C–O symmetric stretch [50]											
(19)	1451–1469	1462 → Poly A [39] 1463 → N1–H (cytosine), C4–C5 (cytosine), CH [40]	√	√	√	√	√	√	√	√	√	√	√
(20)	1545–1555	1545 → C6–H deformation mode [40]			√	√	√	√					
(21)	1574–1578	1577 → Purine bands mainly G contribution, base ring modes, mainly G + A (increase upon methylation) [51,52] 1580 → G, A from nuclei of MCF-7 cells [53]			√	√					√		
(22)	1603	1603 → Cytosine (NH2) [53]							√	√			
(23)	1643–1655	1655 → T, G, C (ring breathing modes of the DNA/RNA bases [51]					√	√					
(24)	2883–2891	2889 → CH_2_ asymmetric stretch of lipids and proteins [43]	√	√	√	√	√	√	√	√	√	√	√
(25)	2918–2950	2940 → CH_2_ asymmetric stretch [43]	√	√	√	√	√	√	√	√	√	√	√

## Data Availability

The datasets used and/or analyzed during the current study are available from the corresponding author on reasonable request.

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
