# Peer review of "Label-Free Human Disease Characterization through Circulating Cell-Free DNA Analysis Using Raman Spectroscopy"

_ijms, 2023, doi:10.3390/ijms241512384_

Round 1

Reviewer 1 Report

Reviewers comments

1. Add line numbers to the manuscript

2. Page 2: Put a relevant reference that matches ccfDNA Raman spectra. If not found, atleast add a similar reference of similar types of DNA

3. While comparing, please Arrange different Raman spectra of DNAs in a 'stacked' mode, so that the changes in peaks be distinguished. 

4. Elaborately mention molecular and structural differences among the DNAs to explain the reasons behind the origin of different peaks at the Raman spectra of different DNAs.

5. Add a 'conclusion' section to the manuscript and include the key findings of this research article.

Reviewer 2 Report

The authors report Raman spectra of circulating cell free DNA (ccfDNA) from various samples under different conditions. Though the work is interesting, I am afraid there are many questions that must be answered and the overall quality of the manuscript must be improved.

1)      The authors report differences in the ccfDNA of male and females. Especially the bands at 914 and 963 cm-1. Is the difference really significant? Is there any reference to show that ccfDNA differs based on gender? Moreover, the authors speculate that the difference is due to Y chromosome without any basis.

2)      Supplementary figure 2 shows crystalline structures. The authors mentioned that they randomly measured 5 different points and averaged. However based on the crystalline structures, spectra may be quite different. It is important to show the standard deviation for each of the averaged spectra for clarity.

3)      The authors report the appearance of Adenine related peak only in MCF12A DNA. Do the authors mean that there is no Adenine in all other DNA samples?

4)      The authors claim that the peak around 2100 cm-1 in Raman spectrum of MCF12A-ccfDNA in Figure 2 is from inorganic substances. I am not sure what they mean by that. If it is introduced during the extraction process, it should have been present in other conditions too. However, it is not observed in MCF7-ccfDNA in Figure 3.

5)      Authors show that there is no difference in Raman spectrum of ccfDNA before and after therapy. Given the situation, I am afraid Raman spectroscopy may not be appropriate to evaluate various conditions.

6)      Are there differences in Raman spectrum of ccfDNA from patient to patient? Such a comparison is necessary.

7)      Colors used for traces are hard to see. Figures must be made in a way so that spectra are visible clearly.

8)      What is the laser power at the sample point?

9)      Was the sample prepared on metallic plate or CaF2 window?

Overall quality of English language is good. Only minor editing of English language is required.

Round 2

Reviewer 2 Report

The authors partially addressed raised concerns.

none